# Instant classification for the spatially-coded BCI

**Alexander Maÿe[1]\*, Raika Rauterberg[1,2], Andreas K. Engel[1]**

**1** Department of Neurophysiology and Pathophysiology, University Medical Center Hamburg-Eppendorf, Hamburg, Germany, **2** Department of Biomechanics, Technical University Hamburg, Hamburg, Germany

\* a.maye@uke.de

## Abstract

The spatially-coded SSVEP BCI exploits changes in the topography of the steady-state visual evoked response to visual flicker stimulation in the extrafoveal field of view. In contrast to frequency-coded SSVEP BCIs, the operator does not gaze into any flickering lights; therefore, this paradigm can reduce visual fatigue. Other advantages include high classification accuracies and a simplified stimulation setup. Previous studies of the paradigm used stimulation intervals of a fixed duration. For frequency-coded SSVEP BCIs, it has been shown that dynamically adjusting the trial duration can increase the system's information transfer rate (ITR). We therefore investigated whether a similar increase could be achieved for spatially-coded BCIs by applying dynamic stopping methods. To this end we introduced a new stopping criterion which combines the likelihood of the classification result and its stability across larger data windows. Whereas the BCI achieved an average ITR of 28.4±6.4 bits/min with fixed intervals, dynamic intervals increased the performance to 81.1±44.4 bits/min. Users were able to maintain performance up to 60 minutes of continuous operation. We suggest that the dynamic response time might have worked as a kind of temporal feedback which allowed operators to optimize their brain signals and compensate fatigue.

**Data Availability Statement:** All data files are available from the Zenodo database (URL: https://doi.org/10.5281/zenodo.5119527).

**Funding:** The work described in this paper was supported by the German research Foundation (DFG www.dfg.de) through project TRR 169/B1.

## Introduction

In current research on brain-computer interfaces (BCIs), the optimal trial duration or data size for recognizing user intent is typically derived from an offline analysis of data recorded from several participants of a study. The optimization finds a trade-off between longer trial durations, in which more data can be recorded and thereby classification accuracy increases, and shorter trials, which increase the number of commands per unit of time. Information transfer rate is a derived performance indicator for BCIs which combines accuracy and duration and therefore is frequently used for the objective function. But of course all other performance measures that have time in the denominator, like output characters per minute or utility [1], increase with shorter trials. In general, the optimal parameters are fixed for each participant or even the studied population after the training phase, but whether or not they are also optimal for the application or online phase is less frequently considered. Fatigue, concentration lapses and other factors affect the brain signal quality, which is why a more fine-grained

The funders had no role in study design, data collection and analysis, decision to publish, or preparation of the manuscript.

**Competing interests:** The authors have declared that no competing interests exist.

or dynamic adaptation of trial duration would be desirable for achieving short reaction latencies of a BCI.

The abundance of sophisticated methods for improving classification accuracy notwithstanding, reducing the amount of data is an important approach for increasing a BCI's output rate. Compared to the effort that is being made towards increasing accuracies, surprisingly little research is dedicated to reducing the amount of data needed for classification and hence trial duration. A systematic review of the few existing methods for a "dynamic stopping" of data recording to maximize ITR in P300-based BCIs is given in [2]. The evaluation showed that all methods improved performance by factors of 2 to 4. Almost all of them were robust in the sense that even in the worst case, the performance of the BCI did not drop below that for fixed trial durations.

More recently, several dynamic stopping (DS) approaches for BCIs which employ the steady-state visual evoked potential (SSVEP) have been developed. One method is to stop data recording when the SSVEP response strength reaches a threshold [3]. Correlation with a sinusoidal template signal is used as an indicator of response strength, and the correlation strength for the trial duration where ITR peaks (in a calibration data set) is taken as the threshold. Another idea is to output a decision when the class-conditional probabilities reach a given threshold [4]. As classification frequently relies on correlations between class-specific templates and the observed data, the difference of the strongest correlation and the second-best match can be taken as a certainty measure, and a decision can be made when it exceeds a threshold [5]. The threshold can be dynamically determined by converting features of the EEG signal to target probabilities through the softmax function and using them to weigh the cost of collecting more data up against the certainty of a correct classification [6]. Based on a model for the distribution of features when the user is attending to the target and to non-targets, Bayes' theorem can be used to calculate the posterior probability for a correct decision, and online data recording can be stopped when it crosses a threshold [7]. The thresholds can be specific for each target and determined by the separability of the target and non-target feature distributions [8]. What all these approaches have in common is that the computational overhead of such methods is negligible; therefore, it is somewhat surprising that DS methods did not become a standard component of current BCI paradigms.

In SSVEP-based as well as P300-based BCIs, stimulation is cyclic: a brightness or color contrast is switched back and forth to generate a flicker in the former, and a row/column combination is highlighted to elicit a P300 response in the latter. It seems obvious that the most useful time points for evaluating whether a classification can be made are when another cycle in the stimulation is finished. Hence the cycle duration determines the temporal granularity of the dynamic stopping. Lower cycle times or, correspondingly, higher stimulation frequencies afford finer granularity and hence better adaptivity to the signal quality and faster response of the BCI. On the other hand, higher stimulation frequencies are known to elicit weaker SSVEP responses [9–11]. We therefore wanted to explore whether and to which extent shorter stimulation cycles can reduce the BCI's latency and increase its ITR. To this end, we used a flicker stimulus of 60 Hz. Compared to the alpha and beta frequency bands which are classically used in SSVEP BCIs, neuronal background activity at this frequency is lower, resulting in a similar signal-to-noise ratio and classification accuracy [11, 12]. Since higher stimulation frequencies generally cause less visual fatigue [10, 13], their widespread application in SSVEP BCIs could help improve user comfort.

The second innovative aspect of our investigation is the introduction of DS in a BCI paradigm which employs the different topographies of the response evoked by a single flicker that appears at different locations in the visual field [14]. Whereas in the majority of SSVEP BCIs the control channels are encoded in different frequencies and/or phases of a set of flicker

stimuli, in this approach different spatial positions relative to a single flicker define the control targets. One important advantage of this approach is that, unlike in frequency-coded SSVEP BCIs in which the user has to gaze at the flicker stimulus, in our spatially-coded SSVEP BCI, the flicker always appears in the extrafoveal field. We have argued that this property likely has advantages with respect to visual fatigue [15]. In addition, the high stimulation frequency that increases the temporal granularity of the dynamic stopping at the same time can also be expected to improve user comfort.

## Materials and methods

### Participants and EEG recording

Fourteen subjects participated in the study. They were between 21 and 57 years old (mean: 30), and 4 of them were females. Ten subjects had previously participated in BCI studies, and two subjects were among the authors of this study. All of them had normal vision and were free of neurological and ophthalmological disorders. The study was approved by the ethics committee of the medical association of the city of Hamburg, Germany. Informed consent was signed by all participants before commencing the experiment.

The experiment took place in a regular lab environment with ambient illumination from ceiling lights and without any electrical or acoustic shielding. EEG was recorded using 32 active electrodes placed according to the 10/20 international system and an ActiveTwo AD-box amplifier (BioSemi Instrumentation, Amsterdam, The Netherlands). The sample rate was 2048 Hz. The labstreaminglayer (https://github.com/sccn/labstreaminglayer/) was used to synchronize EEG data with triggers, write data to a file or stream them into the online processing.

### Stimulation and experimental procedure

The stimulation was based on a previous study in which we introduced the concept of the spatially-coded SSVEP BCI [14]. A large disc in the center of the screen (19° visual angle) provided the flicker by flipping from black to white and vice versa, and the 5 targets were arranged on top, below, to the left, to the right and in the center of the disc (see Fig 1). It is important to note that the targets that the user fixated in order to select the associated command were not flickering. Therefore, the static target in the center of the disc effectively made the flickering area an annulus. Matlab (The Mathworks, Natick, MA, USA) and the Psychophysics Toolbox [16–18]were used to generate the stimulation and to control the experiment. We used an EIZO FlexScan F931 CRT monitor at 120 Hz refresh rate for displaying the stimulus, and the viewing distance was 50 cm.

Participants were instructed to gaze at all five targets in ascending order. The target to be attended was cued for two seconds before fixating it for four seconds. After fixating the five

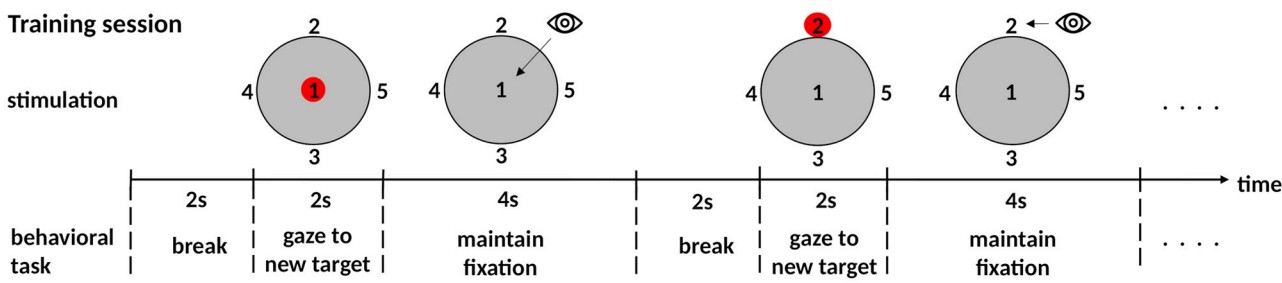

**Fig 1. Schema of the visual stimulation and the procedure in the training session.**

targets in sequence, there was a break of two seconds before the first target of the next sequence was highlighted. Prior to every new sequence, the numbers 1-5 were randomly assigned to the target positions. For the training, the numeric values were not relevant, because the target position to be focused was cued. In the online session, however, there were no cues, and participants had to gaze at targets in ascending numeric order. Hence the randomized assignment during the training familiarized participants with the procedure of the online session.

Before starting the experiment, participants were given the opportunity to explore the stimulation and ask questions. They were requested to avoid head movements, eye blinks, or swallowing during a sequence; however, they could do so during the breaks between sequences. The training session comprised 30 sequences. The resulting training data was pre-processed and used to train the online classifier.

In the subsequent online session, subjects were instructed to gaze at all five numbers in ascending order. Again, the numbers were distributed randomly after each sequence. This time, the participants had 5 seconds at the beginning of each sequence to memorize the new positions of all numbers. Afterwards, they should keep their gaze fixed at each target until an audible signal triggered them to look at the next one. The pitch of the beep provided feedback on whether the corresponding classification was successful or not. Following the beep, subjects had half a second to adjust their gaze to the next target. The trial length was not fixed in the online session. Instead, the dynamic stopping algorithm (see below) started the next trial whenever a sufficiently reliable decision for the current data window could be made. A schematic of the online task is shown in Fig 2.

After each sequence, the time needed to complete the five trials was displayed. This feedback should encourage participants to focus and swivel their gaze as swiftly as possible. Five sequences formed one block. Within a block, all sequences followed each other with a five-second break at the beginning to memorize the new number positions. The online session comprised at least 10 blocks. After the tenth block, the participant could continue the experiment *ad libitum* and try to improve the own performance.

## Data analysis

Data were filtered by a zero-phase finite impulse response filter with a 55-65 Hz pass band. The optimal trade-off between low filter orders enabling smaller data windows and high ITRs was determined on the training data from each participant by grid search.

A standard canonical correlation analysis (CCA [19]) was then employed to calculate correlations with a reference signal to be used as features for the classification. A detailed description of the classification method is given in [14], but the main idea is reproduced here. CCA

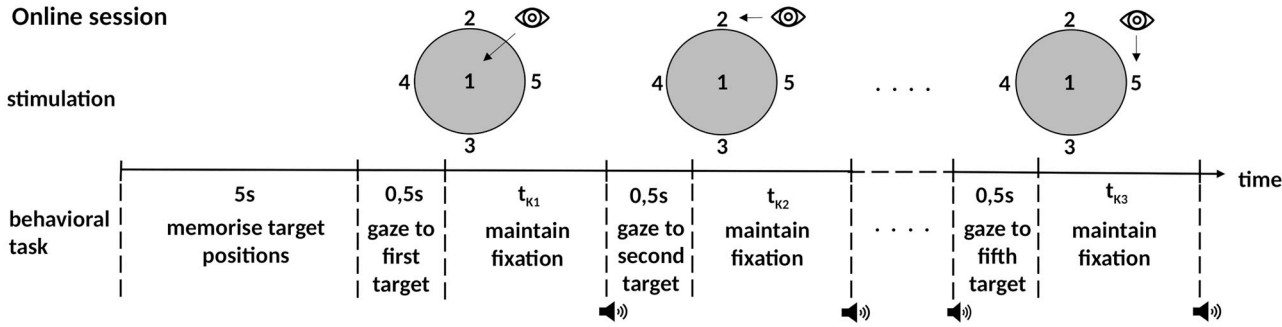

**Fig 2. Schema of the visual stimulation and the procedure in the online session.**

determines spatial filters $\mathbf{A}$ and $\mathbf{B}$ such that the set of correlations $\mathbf{r}$ between two multi-variate signals $\mathbf{X}$ and $\mathbf{Y}$ is maximized:

$$\mathbf{r} = \max_{A,B} \text{corr}(\mathbf{AX}, \mathbf{BY})$$
(1)

Here, $\mathbf{X} \in \mathbb{R}^{L \times T}$ are $T$ samples of an $L$-channel EEG signal, and $\mathbf{Y}$ is a sinusoidal reference signal at the frequency of the flicker stimulation

$$\mathbf{Y} = \begin{bmatrix} \sin(2\pi f_{stim} t) \\ \cos(2\pi f_{stim} t) \end{bmatrix}$$
(2)

with $t = [0 \ldots T/f_{sample}]$ in steps of the inverse of the sampling frequency $f_{sample}$. The `canon-corr` function in Matlab solves Eq 1 efficiently by QR and singular value decomposition.

For each of the $C = 5$ targets, the corresponding trials from the training session were concatenated along the time dimension, and the class-specific filters $\mathbf{A}_c$ and $\mathbf{B}_c$ were determined. Then each trial was filtered with all the $c = 1 \ldots C$ filters, and the resulting canonical correlations were merged into the feature vector $f$ for the trial. Together with the corresponding class labels, these feature vectors comprised the training data for the LDA classifier. This classification method is frequently used in BCI applications, and it performed reasonably well in our previous studies [14, 15].

LDA [20] classifies a feature vector $f$ according to the maximum posterior probability of $f$ belonging to any of the $c$ classes:

$$\hat{c}(f) = \underset{c}{\text{argmax}} P(c|f).$$
(3)

Using Bayes' theorem, the posterior probability can be calculated by:

$$P(c|f) = \frac{P(f|c)p(c)}{P(f)} = \frac{P(f|c)p(c)}{\sum_c P(f|c)p(c)}.$$
(4)

where $p(c)$ is the prior probability of class $c$. The class-conditional probability distribution $P(f|c)$ is estimated using $d$-dimensional multi-variate Gausians:

$$P(f|c) = \frac{1}{(2\pi)^{d/2}|\Sigma|^{1/2}} \exp\left(-\frac{1}{2}(f - \mu_c)^T \Sigma^{-1}(f - \mu_c)\right)$$
(5)

with the class-centroids $\mu_c$ and the covariance matrix $\Sigma$ calculated from the training data. The implementation of LDA in Matlab's `classify` function was used.

To get an estimate of the classification accuracy on the training data, a leave-one-sample-out cross-validation approach was employed. For each trial in the training data, the procedure for calculating feature vectors was repeated, but the respective trial was excluded. The classifier outputs for the excluded trials were used to estimate the offline classification accuracy. ITR was then calculated by:

$$ITR = \frac{60}{T}\left(\log_2 C + G\log_2 G + (1-G)\log_2 \frac{1-G}{C-1}\right)$$
(6)

where $G$ is the classification accuracy, $C$ the number of classes and $T$ the time window [21–23]. The optimal fixed data length was determined by varying the data length and finding the maximum of the ITR for each participant.

## Classification with dynamic time windows

The critical component of any dynamic stopping method is a strategy for determining the reliability of a classification result. We adopted the simple idea that consecutive classifications on an increasingly larger data window should return the same result. This heuristic has been successfully used in a P300-based BCI spelling application where it increased the bit rate by 20% on average [24].

We found that performance could be further improved by combining it with a threshold on the posterior probability $P(c|f)$ of the winning class from the LDA classifier (Eq 4). This is the equivalent of the Bayes criterion for DS in SSVEP-BCIs that use the magnitude of the correlation coefficients for classification [7, 8, 25]. Hence the classification result is final when a fixed number of $N$ classifications in succession have each yielded the same result with at least a minimum posterior probability of $P$. The optimal combination of parameters $N$ and $P$ for which the BCI shows the highest ITR were determined by grid search. A schematic of the DS algorithm is shown in Fig 3.

The update frequency for evaluating the stopping criterion in principle is only limited by the sampling rate of the EEG amplifier and the computing power of the machine running the DS algorithm. It turned out, however, that the software interface to the amplifier returned EEG data always in chunks that were multiples of 131 samples. At a sampling rate of 2048 Hz, this corresponds to a maximum update frequency of 15.6 Hz and limits the temporal resolution of the dynamically adjusted trial duration to about 64 ms. The frequency of 60 Hz for the visual stimulation would enable a resolution of 16.7 ms, but the technical setup did not permit this advantage to play out in our study. Dedicated amplifiers sending data packets as frequently as every 4 ms have been developed [8].

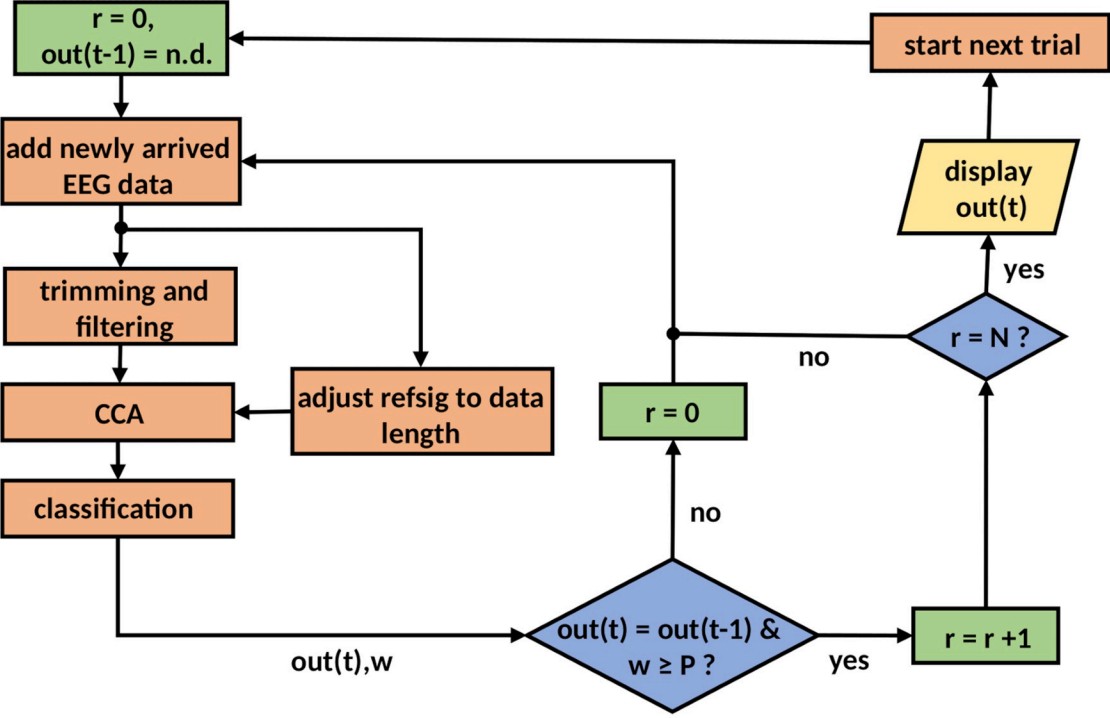

**Fig 3. Flow diagram of the dynamic stopping algorithm.**

In our DS algorithm, the feature vectors to be classified were calculated from data windows which were growing with every iteration. We found that classification accuracy was compromised when providing the classifier with the raw training data having a fixed trial length of 4 seconds. The issue was solved by trimming the training data to the same length as the data to be classified. Also, the spatial filters **A** and **B** as well as the training feature vectors were re-calculated for each data length.

## Results

### Classification time and accuracy

We first analyzed the relation between the size of the data window and the accuracy of the classification. As expected, larger windows entailed higher classification accuracies (Fig 4A). Beyond around 2 s, the increase became marginal though, and the average offline accuracy approached 93%. The corresponding ITRs peak at data windows of 1 s or less for all participants but one (Fig 4B).

We then analyzed the online performance of the DS classification for each participant. Fig 5 shows the distribution of ITRs in the online session. The average ITRs that participants reached across blocks range from 21.6 bits/min to 166.3 bits/min. Peak performance in a single block was at 262.8 bits/min, and the lowest performance was 9.29 bits/min. Whereas classification accuracy was higher in the training session on average ($0.93\pm0.08$ vs. $0.85\pm0.08$, paired Student's t-test $p = 1 \times 10^{-5}$), the early stopping in the online session yielded a substantially increased ITR ($28.39\pm6.4$ bits/min vs. $81.04\pm44.53$ bits/min, $p = 2.6 \times 10^{-4}$).

We also optimized the data window size to maximize ITR on the training data, and we used this window to estimate the ITR that we would have observed in the online experiment if a fixed data window had been used. This was possible because DS resulted in data windows that typically were larger than the optimal fixed size. The comparison between DS and the optimal fixed window in Fig 5 suggests that DS improved the ITR for most participants; in 6 of them this improvement reached a statistical significance of 0.05. Only for two participant the optimal fixed window performed significantly better than DS. Numerical results for the DS method are listed in Table 1.

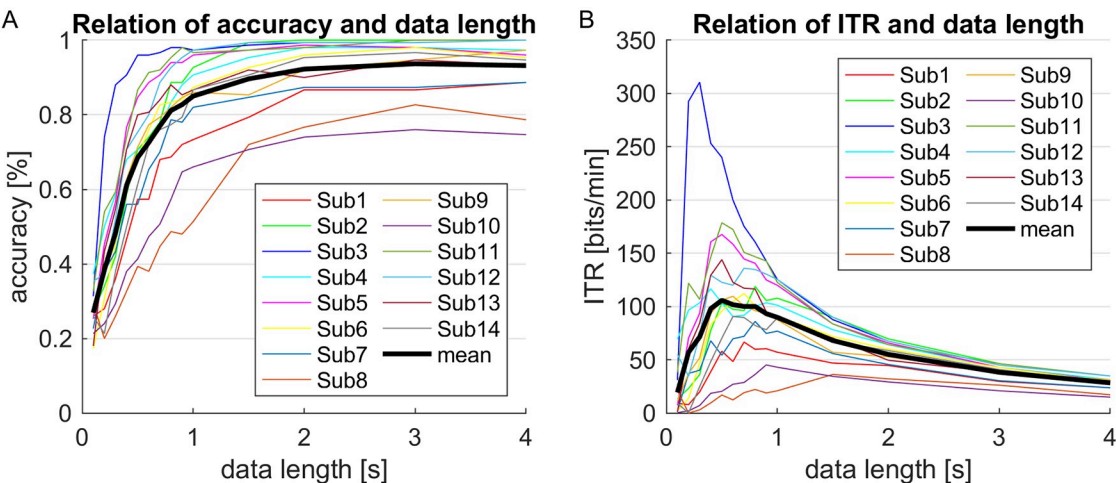

**Fig 4. Effect of the data window size.** A: Classification accuracy. B: ITR.

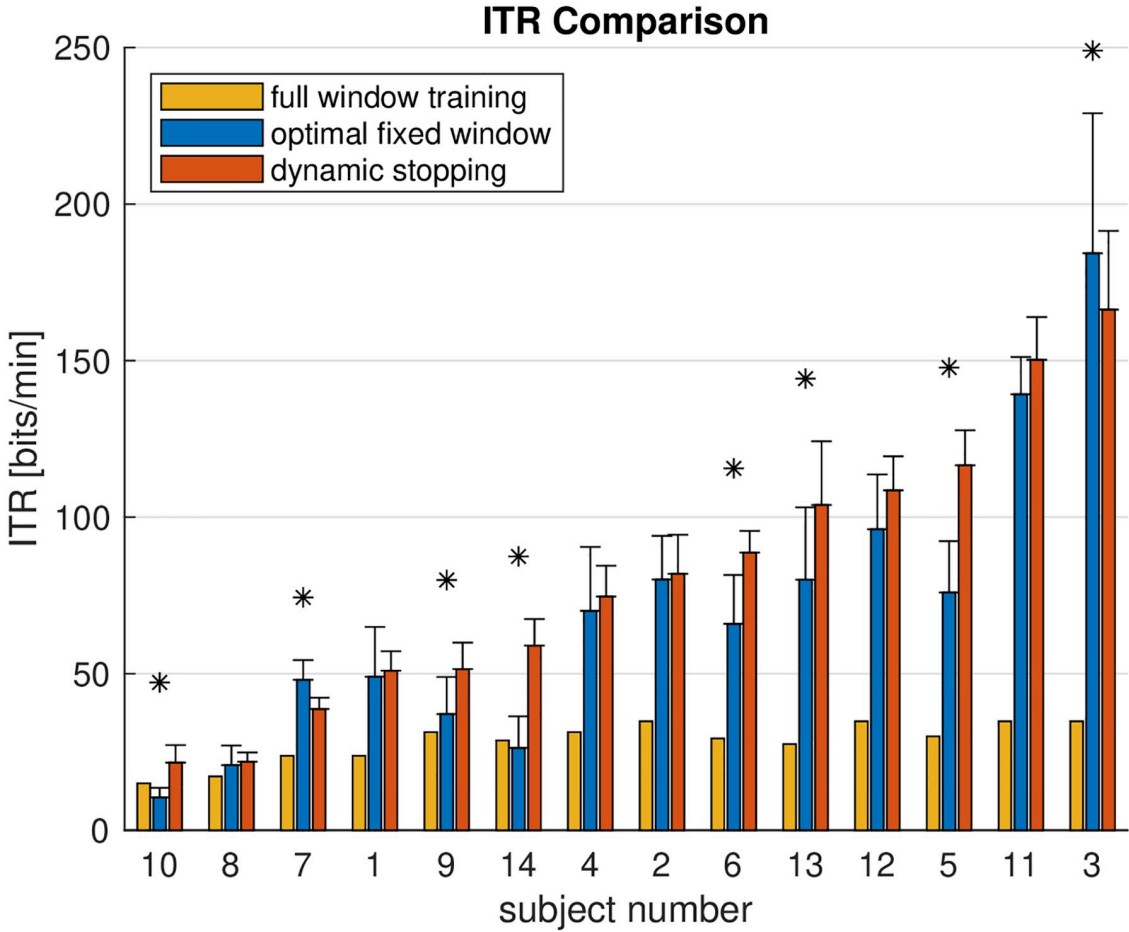

**Fig 5. Online and offline performance (ITR) of the participants.** Bars display the mean ITR across the 10 or more blocks that each participant completed; whiskers show the standard deviation. Asterisks mark differences of the ITRs for the optimal fixed window size and dynamic stopping at the 0.05 significance level (paired Student's t-test).

The average size of the dynamic window in the online session was between 1 and 1.5 s for all five targets. Target 3, at the bottom of the flicker stimulus, had the longest average window size and the largest variance; target 1, in the center, required shorter windows and was recognized with the lowest variance in window size (Fig 6A). We reckon that this is related to the size of the flicker in the visual field when the participant gazed at the respective target. This size was small when gazing at the bottom and large when gazing at the center of the stimulus, thus stimulating larger areas on the retina and in the cortex and leading to stronger responses. In addition, the SSVEP response seems to be stronger for stimuli in the lower visual field than in the upper [11]. This, however, does not seem to impact the classification accuracy, which was similar between targets 3 and 1 with a few more errors for the latter (Fig 6B).

### Effect of the SSVEP response magnitude

The performance variation across participants that is evident from Fig 5 prompted us to investigate whether it bears a relation to the magnitude of the SSVEP response. We quantified the response strength by the ratio of the EEG power at the stimulation frequency and the average

**Table 1. Performance of the dynamic classification for each subject.**

| Subject | Classif. time [s] | Accuracy online | Accuracy offline | ITR [bits/min] |
|---|---|---|---|---|
| 1 | 1.3 | 0.77 | 0.89 | 50.95 |
| 2 | 1.3 | 0.91 | 1.00 | 81.91 |
| 3 | 0.7 | 0.91 | 1.00 | 166.30 |
| 4 | 1.4 | 0.91 | 0.97 | 74.66 |
| 5 | 1.0 | 0.94 | 0.96 | 116.56 |
| 6 | 1.3 | 0.94 | 0.95 | 88.71 |
| 7 | 1.9 | 0.80 | 0.89 | 38.76 |
| 8 | 2.8 | 0.75 | 0.79 | 22.78 |
| 9 | 1.9 | 0.88 | 0.97 | 51.50 |
| 10 | 2.1 | 0.66 | 0.75 | 21.61 |
| 11 | 0.7 | 0.92 | 1.00 | 150.26 |
| 12 | 0.8 | 0.86 | 1.00 | 108.58 |
| 13 | 0.9 | 0.86 | 0.93 | 103.91 |
| 14 | 1.2 | 0.79 | 0.95 | 59.45 |
| Mean | 1.4 | 0.90 | 0.93 | 81.14 |
| Std. dev. | 0.6 | 0.08 | 0.08 | 44.43 |

power in the interval from 58.75 Hz to 61.25 Hz excluding 60 Hz (signal-to-noise ratio, SNR). We found indeed that the median ITR is directly proportional to the response strength ($r = 0.79$, $p = $ 7e-4, Fig 7).

## Performance over time

Every participant completed at least 10 blocks in the online session. This allowed us to assess the stability of the performance across the time of usage. To this end, we calculated the percentage of change of the ITR over the course of the online session relative to the ITR from the first online block. Fig 8 suggests that there is a weak trend towards higher ITRs over time.

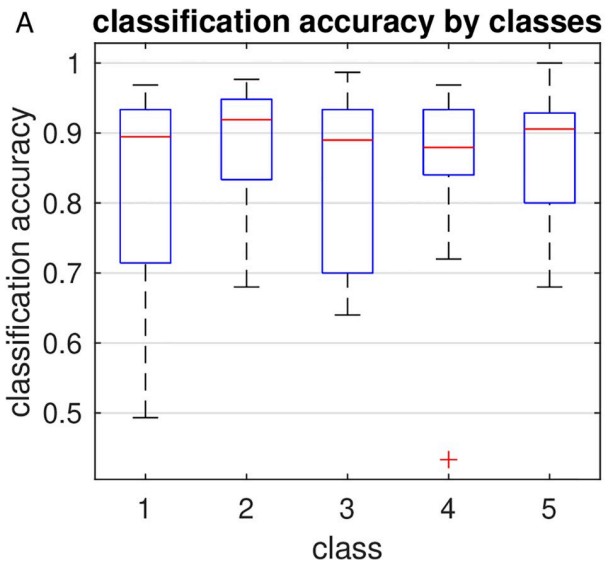
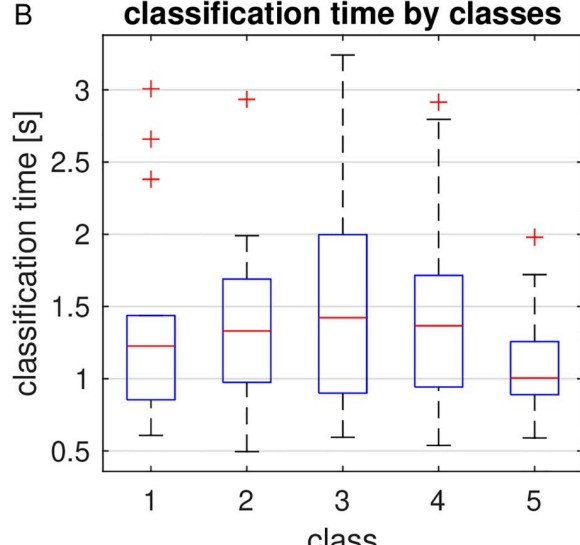

**Fig 6. Classification accuracy (A) and trial duration (B) for each target.**

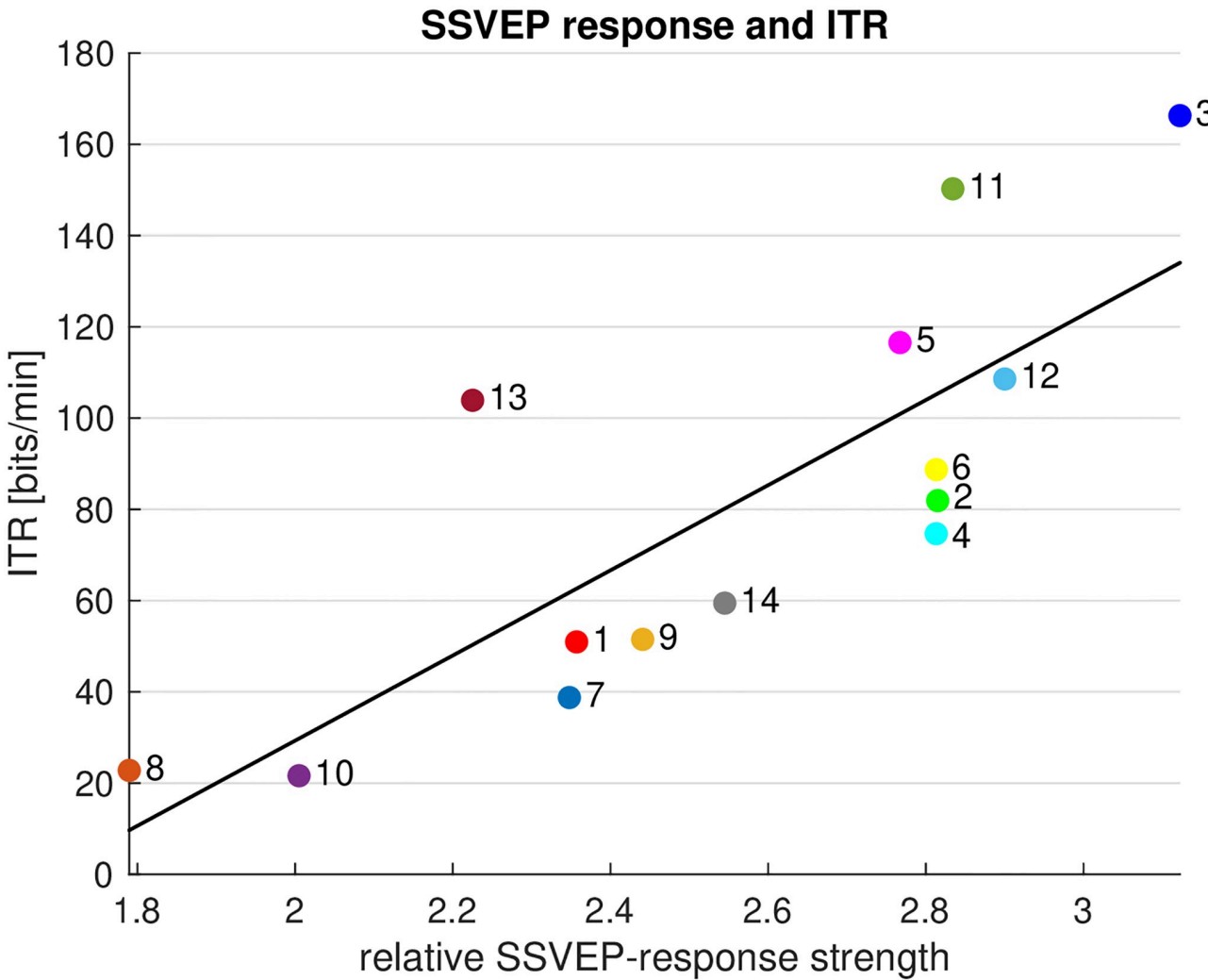

**Fig 7. Relation between SSVEP response magnitude (on the abscissa) and ITR (on the ordinate) for all participants (numbered dots).**

## Analysis of meta parameters

The variable data length required some considerations concerning the bandpass filtering of the raw data. The filter order of the bandpass filter constrains the minimum data length. In order to classify short data windows without deteriorating accuracy caused by ineffective filtering, we searched for an optimum between a low filter order and high ITR values. To this end, we recorded additional data sets from subjects 10–14, who later also participated in the study. Analyzing these data sets, we found that a filter order of 66 yielded good results on average (Fig 9).

To make the dynamic stopping as effective as possible, we searched for an optimum combination of the two stopping parameters: the minimum reliability of a classification $P$, calculated in the discriminant analysis, and the number of consecutive classifications leading to the same result $N$. We analyzed the ITR as a two-dimensional function of these two parameters. Based on Fig 10, $P = 0.95$ and $N = 2$ were chosen for all subjects.

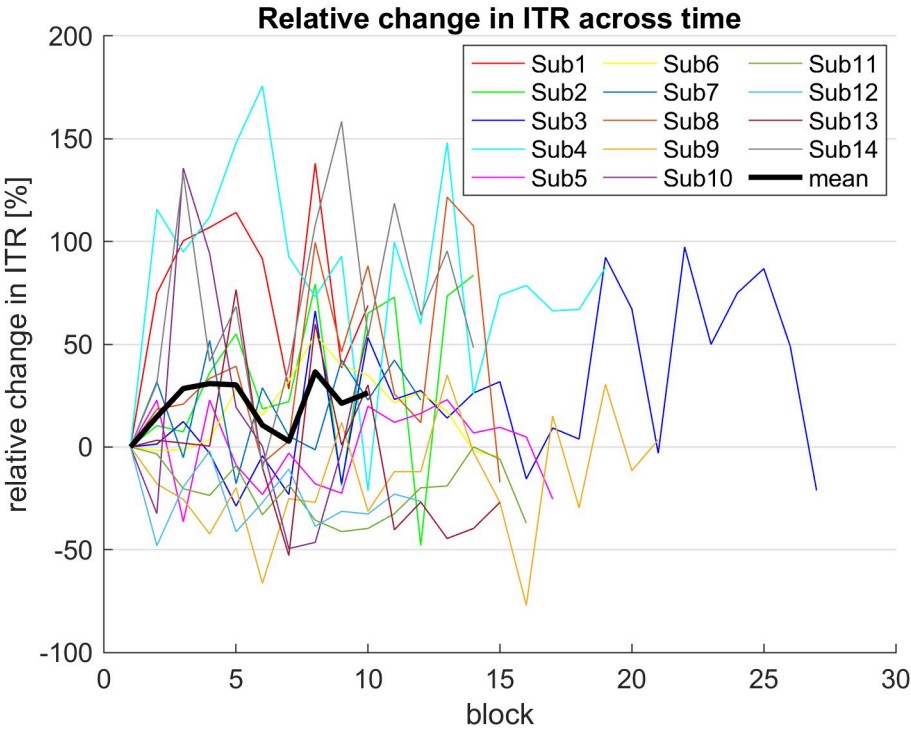

**Fig 8. ITR change over the online session w.r.t. the first online block for each participant.** Each subject completed a minimum of 10 blocks and continued thereafter at their own discretion.

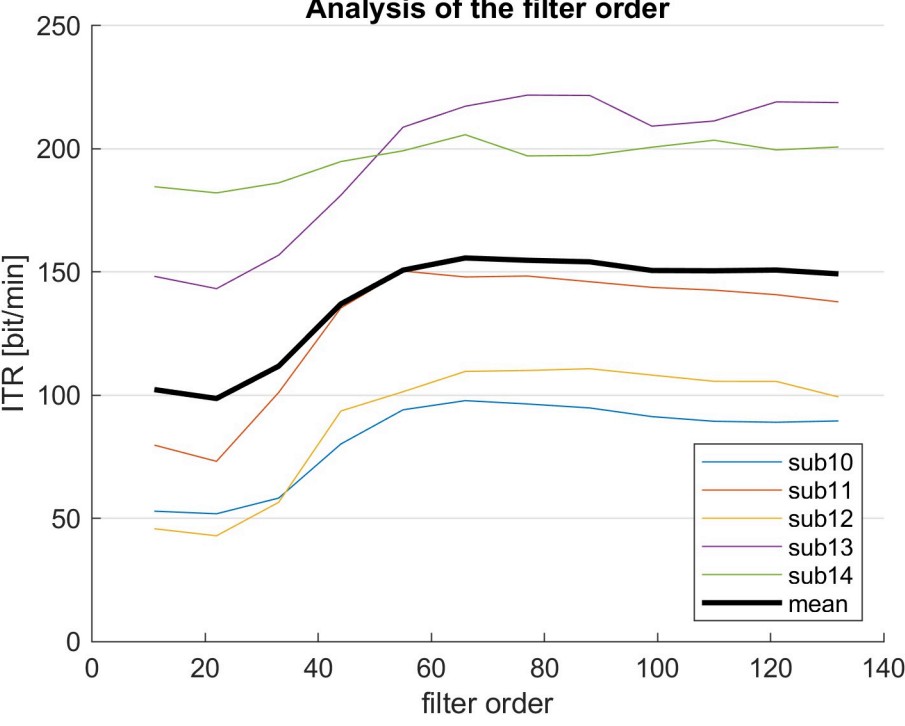

**Fig 9. Effect of filter order on ITR.**

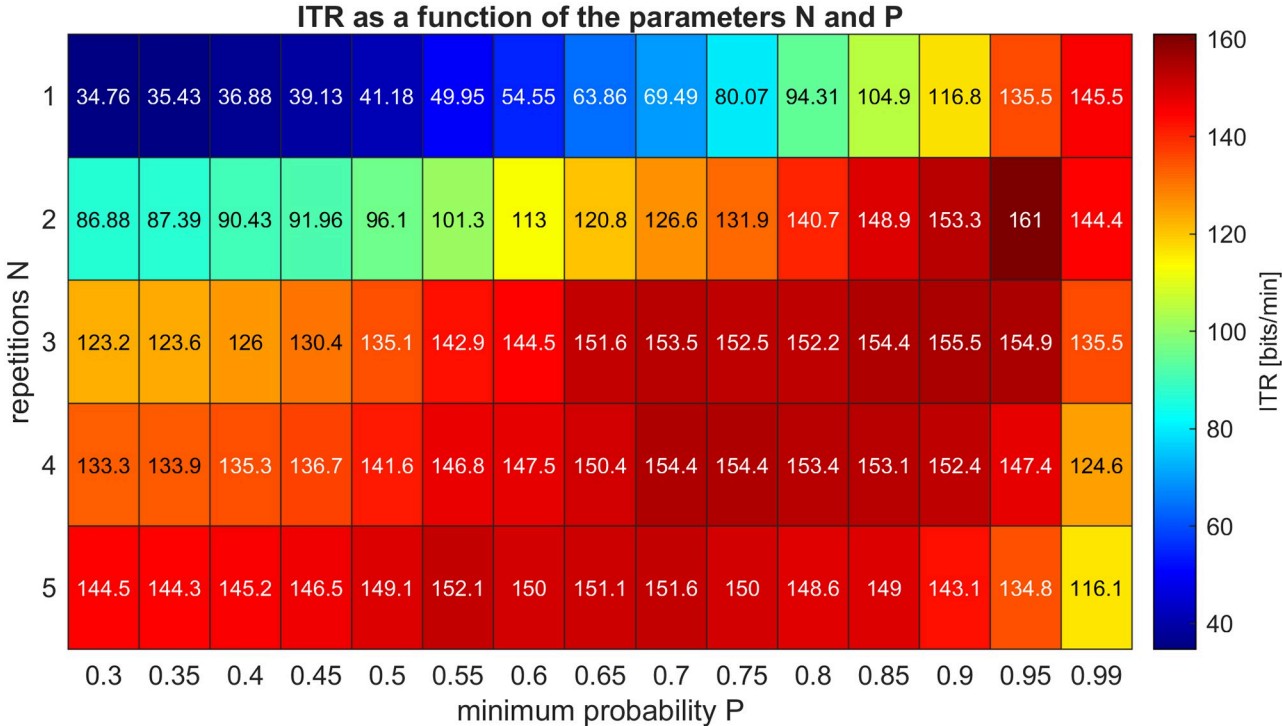

**Fig 10. Influence of parameters N and P of the dynamic stopping on the ITR.**

## Discussion

The proposed DS method yielded a performance improvement for the majority of the participants in the study. For 6 of the 14 participants, the ITR showed a systematic increase in comparison to a fixed trial duration. In another 6 participants, there was no or only a small improvement. ITR was lower for dynamically adjusted trial durations in the remaining two participants.

A comparison with related studies on dynamic stopping methods for frequency-coded SSVEP BCI shows that the spatially-coded SSVEP BCI gains the largest relative increase in ITR when the trial duration is dynamic (see Table 2). Attention should be paid, however, to the method for evaluating the performance for the fixed optimal trial length. Some studies use a training data set for finding the optimal trial length and then evaluate the performance on the same data, likely leading to an overestimation of the performance. In contrast, we determined the optimal fixed trial duration from the training data, and we evaluated it on the same data

**Table 2. Comparison of ITR with related studies.**

| Reference | ITR (bit/min) | | Improvement (%) |
|---|---|---|---|
| | **Fixed length** | **Dynamic length** | |
| [3] | 37.71 | 41.08 | 8.94 |
| [5] | – | 57.3 | – |
| [6] | 134.25 | 164.72 | 22.7 |
| [7] | 239.6 | 257.6 | 7.5 |
| [8] | 300.7 | 330.4 | 9.9 |
| our study | 63.44 | 81.14 | 27.9 |

like our DS method, i.e., the online data. This was possible because the dynamic trial durations were typically a little longer than the fixed optimal length.

The observed classification accuracies of over 80% for trials of 1 s or longer confirm the finding in earlier studies that high flicker frequencies still afford reliable classification of the SSVEP response despite lower response amplitudes [10–13, 26]. Across the participants of the study, performance ranged from about 22 to 166 bits/min. In order to find possible causes of this substantial variation, the relation between ITR and SSVEP response strength was analyzed. We found a significant correlation between these parameters. Thus, the effectiveness of the DS depends on the individual SNR of the SSVEP response. To remedy the performance variation, we suggest running a quick test of the SSVEP SNR before using the BCI and adjust the flicker frequency as appropriate. In general, if more SSVEP BCI studies would employ high-frequency stimulation, BCI researchers could glean a better overview of the response properties across the population.

As mentioned before, the average trial duration of 1-1.5 s (see Fig 6A) was typically longer than the optimal fixed window size of 1 s or below (cf. Fig 4). This may indicate that our stopping criterion is rather conservative and has room for improvement. Whereas the average trial duration of the DS method was calculated from online data, the optimal fixed window size however was estimated from training data which were recorded with a fixed trial duration of 4 s. Hence the difference may also result from a generally lower classification accuracy in the online session incurred by the tighter timing of the dynamic regime. For one thing, memorizing the target locations and focusing them in a sequence during the online session may have been more challenging than just gazing at the cued location and switching to the next at a low pace during the training. For another, the exertion caused by trying to improve the own performance in each new block might have degraded the EEG signal quality in the online session in comparison to that in the training session.

Despite the challenges of the fast gaze changes in the online session, we observed no deterioration of the performance over the usage time of the BCI. There are two factors which might have influenced the performance over time. On the one hand, a training effect which was fueled by the auditory feedback and the participants' inducement to reach their personal minimum time might have caused an increase of the BCI performance. On the other hand, the participants also experienced fatigue over the course of the online experiment. It seems that both effects roughly balanced out.

With respect to a potential training effect in the application phase we would like to point out that the online session may implicitly have worked as a neurofeedback system. The auditory feedback informed the participant whether the command was correctly recognized or not. In addition, DS generated feedback through the recognition latency. When trial duration is dynamically adjusted, better EEG signal quality allows the classifier to use shorter data segments and hence output the command faster. Participants may have experienced the joint accuracy and latency feedback as rewarding, and it might have trained them to generate EEG signals with better discriminability. Therefore, it would be interesting to investigate the behavior of performance over longer usage times.

Further improvements may be achieved by a more detailed consideration of the meta-parameters of the DS method. In this study, the number of repetitions $N$ and the probability threshold $P$ were fixed for all participants. We have indication though that subject-specific choices can likely improve the performance for some participants. Since we observed a direct relation between the strength of the SSVEP response and the ITR, adjusting the stimulation frequency to the individual response properties could be an additional means to fully exploit the potential of our DS method to improve the performance of the spatially-coded SSVEP BCIs.

## Acknowledgments

The authors would like to express their gratitude to Tiezhi Wang from the University of Oldenburg and Marvin Mutz from the Technical University of Hamburg for supporting the development of the setup and data recording.

## Author Contributions

**Conceptualization:** Alexander Maÿe.

**Funding acquisition:** Andreas K. Engel.

**Investigation:** Alexander Maÿe, Raika Rauterberg.

**Methodology:** Raika Rauterberg.

**Project administration:** Andreas K. Engel.

**Software:** Raika Rauterberg.

**Supervision:** Alexander Maÿe, Andreas K. Engel.

**Visualization:** Raika Rauterberg.

**Writing – original draft:** Alexander Maÿe, Raika Rauterberg.

**Writing – review & editing:** Andreas K. Engel.

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
