## [Decision Letter · Decision Letter 0]

8 Jul 2021

PONE-D-21-13493

Instant classification for the spatially-coded BCI

PLOS ONE

Dear Dr. Maye,

Thank you for submitting your manuscript to PLOS ONE. After careful consideration, we feel that it has merit but does not fully meet PLOS ONE’s publication criteria as it currently stands. Therefore, we invite you to submit a revised version of the manuscript that addresses the points raised during the review process.

We look forward to receiving your revised manuscript.

Kind regards,

Saeed Mian Qaisar, Ph.D.

Academic Editor

PLOS ONE

Additional Editor Comments:

Dear Authors,

Reviewers have now commented on your paper. They are advising that you revise your manuscript. If you are prepared to undertake the work required, I would be please to reconsider my decision.

While submitting the revised version of manuscript, it is recommended that you submit the dataset, used in this study, in .csv or in .mat (MATLAB) format as a complementary zip file.

The reviewer comments can be found at the end of this email or can be accessed online.

Journal Requirements:

4. Please ensure that you refer to Figure 2 in your text as, if accepted, production will need this reference to link the reader to the figure.

Reviewers' comments:

Reviewer's Responses to Questions

**Comments to the Author**

1. Is the manuscript technically sound, and do the data support the conclusions?

Reviewer #1: Yes

2. Has the statistical analysis been performed appropriately and rigorously? 

Reviewer #1: Yes

3. Have the authors made all data underlying the findings in their manuscript fully available?

Reviewer #1: No

4. Is the manuscript presented in an intelligible fashion and written in standard English?

Reviewer #1: Yes

5. Review Comments to the Author

Reviewer #1: I have following questions:

1] The authors should provide the details of canonical correlation analysis (CCA). The authors should also mention and provide the details of software used for analysis and classification (Matlab /Python etc .. and code links)

2] Why authors selected only LDA classifier ?

3] The table mentioning the classification accuracies and other performance measure need to be added.

4] The authors should provide the dataset link.

6. PLOS authors have the option to publish the peer review history of their article (what does this mean?). If published, this will include your full peer review and any attached files.

Reviewer #1: No

---

## [Author Response · Author response to Decision Letter 0]

3 Sep 2021

Thank you very much for reviewing our manuscript. We think the issues raised by the reviewer are appropriate, and we addressed them in the revised manuscript as described below:

1] The authors should provide the details of canonical correlation analysis (CCA). The authors should also mention and provide the details of software used for analysis and classification (Matlab /Python etc .. and code links)

We added a statement about the implementation of the CCA and the classifier we used in our analyses in the section “Material and methods/Data analysis”. 

2] Why authors selected only LDA classifier ?

In the same section, we now give the rationale for using LDA.

3] The table mentioning the classification accuracies and other performance measure need to be added.

A table with numerical values of various performance measures has been added to the section “Results/Classification time and accuracy”.

4] The authors should provide the dataset link.

We uploaded the dataset to the Zenodo repository and provide the link in the submission form.

---

## [Decision Letter · Decision Letter 1]

20 Dec 2021

PONE-D-21-13493R1Instant classification for the spatially-coded BCIPLOS ONE

Dear Dr. Maye,

Thank you for submitting your manuscript to PLOS ONE. After careful consideration, we feel that it has merit but does not fully meet PLOS ONE’s publication criteria as it currently stands. Therefore, we invite you to submit a revised version of the manuscript that addresses the points raised during the review process. Please submit your revised manuscript by Feb 03 2022 11:59PM. If you will need more time than this to complete your revisions, please reply to this message or contact the journal office at plosone@plos.org. Please include the following items when submitting your revised manuscript:A rebuttal letter that responds to each point raised by the academic editor and reviewer(s). You should upload this letter as a separate file labeled 'Response to Reviewers'.A marked-up copy of your manuscript that highlights changes made to the original version. You should upload this as a separate file labeled 'Revised Manuscript with Track Changes'.An unmarked version of your revised paper without tracked changes. You should upload this as a separate file labeled 'Manuscript'.

We look forward to receiving your revised manuscript.

Kind regards,

Saeed Mian Qaisar, Ph.D.

Academic Editor

PLOS ONE

Additional Editor Comments (if provided):

Dear Authors,

Reviewers have now commented on your paper. They are advising that you revise your manuscript. If you are prepared to undertake the work required, I would be please to reconsider my decision.

The reviewer comments can be found at the end of this email or can be accessed online.

Reviewers' comments:

Reviewer's Responses to Questions

**Comments to the Author**

1. If the authors have adequately addressed your comments raised in a previous round of review and you feel that this manuscript is now acceptable for publication, you may indicate that here to bypass the “Comments to the Author” section, enter your conflict of interest statement in the “Confidential to Editor” section, and submit your "Accept" recommendation.

Reviewer #1: All comments have been addressed

Reviewer #2: (No Response)

2. Is the manuscript technically sound, and do the data support the conclusions?

Reviewer #1: Yes

Reviewer #2: Yes

3. Has the statistical analysis been performed appropriately and rigorously? 

Reviewer #1: N/A

Reviewer #2: Yes

4. Have the authors made all data underlying the findings in their manuscript fully available?

Reviewer #1: Yes

Reviewer #2: No

5. Is the manuscript presented in an intelligible fashion and written in standard English?

Reviewer #1: Yes

Reviewer #2: Yes

6. Review Comments to the Author

Reviewer #1: (No Response)

Reviewer #2: Authors proposed spatially-coded SSVEP BCI study. Following comments should consider to improve the quality of the article.

1. The authors failed to show the novelty in their work. Please clarify the innovation in the study.

2. Please re-write the abstract as it looked a very simple one. Abstract should discuss three important things. (i). Limitations of available literature (ii) Method proposed by the author with technical information (iii) advantages and application of proposed method

3. I recommend authors to use multiscale principal component analysis (MSPCA) which is a combination of PCA and wavelets and useful for noise removal from network packets?

The details of MSPCA can be found in “Motor imagery BCI classification based on novel two-dimensional modelling in empirical wavelet transform”

4. For BCI system, signal decomposition methods always play significant role. I recommend authors to have a look on following article

“Motor Imagery EEG Signals Classification Based on Mode Amplitude and Frequency Components Using Empirical Wavelet Transform”

5. Did authors try to use non-linear features for correct identification in BCI? I recommend authors to include discussion of mean energy, mean Teager-Kaiser energy, SHANNON WAVELET ENTROPY and Log energy entropy.

6. The combination of signal decomposition with dimension reduction techniques along with neural networks can be one effective tool for both subject dependent and independent BCI frameworks. Authors need to discuss this issue; detail may be found in “Exploiting dimensionality reduction and neural network techniques for the development of expert brain–computer interfaces”.

7. The authors recorded dataset from very few subjects. Is it possible to collect dataset from more subjects? If it is not possible, at least a discussion is needed for a framework tested on 58 subjects. See following article

“Towards the development of versatile brain-computer interfaces”

8. Please provide a comprehensive comparison of your study with the available literature in terms of classification accuracy, number of channels, features, and execution time with the following articles,

“A new framework for automatic detection of motor and mental imagery EEG signals for robust BCI systems”, “A Matrix Determinant Feature Extraction Approach for Decoding Motor and Mental Imagery EEG in Subject Specific Tasks”, “Motor imagery BCI classification based on novel two-dimensional modelling in empirical wavelet transform”,

“Identification of Motor and Mental Imagery EEG in Two and Multiclass Subject-Dependent Tasks Using Successive Decomposition Index”

9. Please provide the details of future direction and possible solutions to continue this topic.

10. Finally, I suggest authors to sit with English native speaker to improve the writing of proposed work.

7. PLOS authors have the option to publish the peer review history of their article (what does this mean?). If published, this will include your full peer review and any attached files.

Reviewer #1: No

Reviewer #2: No

---

## [Author Response · Author response to Decision Letter 1]

18 Feb 2022

1. The authors failed to show the novelty in their work. Please clarify the innovation in the study.

We made the innovation more explicit in the rewritten Abstract. We think the two innovative aspects also become clear at the end of the Introduction.

2. Please re-write the abstract as it looked a very simple one. Abstract should discuss three important things. (i). Limitations of available literature (ii) Method proposed by the author with technical information (iii) advantages and application of proposed method.

We rewrote the abstract according to the suggested structure.

3. I recommend authors to use multiscale principal component analysis (MSPCA) which is a combination of PCA and wavelets and useful for noise removal from network packets? The details of MSPCA can be found in “Motor imagery BCI classification based on novel two-dimensional modelling in empirical wavelet transform”

We have received some more suggestions for alternative data processing and analysis methods during the revision process of our previous publications on the spatially-coded SSVEP BCI. We explored them all, but none of them outperformed the approach we describe in the manuscript. In addition, we also investigated new analysis methods for SSVEP BCIs like task-related component analysis (TRCA, Tanaka et al., 2013), individual-template CCA (Bin et al., 2011) or convolutional neural networks (Waytowich et al., 2018). For noise removal in SSVEP BCIs, the maximum signal fraction analysis method (MSFA, Wei et al., 2019) has been developed. We found that it can improve the classification accuracy in some circumstances, but that it does not provide an advantage for the system we present in the manuscript. Whereas the application of data processing methods which have shown to be useful in motor-imagery BCIs for SSVEP BCI is certainly an interesting idea, we suggest exploring their potential in a future study.

4. For BCI system, signal decomposition methods always play significant role. I recommend authors to have a look on following article “Motor Imagery EEG Signals Classification Based on Mode Amplitude and Frequency Components Using Empirical Wavelet Transform”

See 5.

5. Did authors try to use non-linear features for correct identification in BCI? I recommend authors to include discussion of mean energy, mean Teager-Kaiser energy, SHANNON WAVELET ENTROPY and Log energy entropy.

The correlation features we used in the study yielded excellent classification results in our previous studies on this paradigm, and similar features are frequently used in other SSVEP BCI systems. We are always interested in evaluating alternative methods which can increase the classification accuracy. The focus of our current manuscript however is on dynamic stopping; therefore, we suggest comparing different methods for feature extraction in a separate study.

6. The combination of signal decomposition with dimension reduction techniques along with neural networks can be one effective tool for both subject dependent and independent BCI frameworks. Authors need to discuss this issue; detail may be found in “Exploiting dimensionality reduction and neural network techniques for the development of expert brain–computer interfaces”.

The recommended article proposes LDA as one of several methods for dimension reduction, and we employ this method in our approach for classification. We would like to point out that the features calculated by eq. (1) are 10-dimensional. Compared to the problem which is analyzed in the recommended article, this already is a rather low-dimensional feature space. Whereas the development of a user-independent version of the spatially-coded SSVEP BCI is an interesting endeavor, it was not the focus of the current study.

7. The authors recorded dataset from very few subjects. Is it possible to collect dataset from more subjects? If it is not possible, at least a discussion is needed for a framework tested on 58 subjects. See following article “Towards the development of versatile brain-computer interfaces”

A main difference between the recommended article and our study is that we ran an online experiment with participants in our laboratory. Hence, we would like to compare the number of participants in our study with related studies on dynamic stopping which also included an online experiment:

Reference in the manuscript Number of participants

3 11

5 12

6 14

7 12

8 12

24 10

25 10

Our study 14

In the light of these numbers, it seems that the size of our cohort is quite standard.

8. Please provide a comprehensive comparison of your study with the available literature in terms of classification accuracy, number of channels, features, and execution time with the following articles,

“A new framework for automatic detection of motor and mental imagery EEG signals for robust BCI systems”, “A Matrix Determinant Feature Extraction Approach for Decoding Motor and Mental Imagery EEG in Subject Specific Tasks”, “Motor imagery BCI classification based on novel two-dimensional modelling in empirical wavelet transform”, “Identification of Motor and Mental Imagery EEG in Two and Multiclass Subject-Dependent Tasks Using Successive Decomposition Index”

All the recommended articles seem to study MI BCIs, which involve different task instructions for the operator and employ other signal analysis methods than SSVEP BCIs. We think, therefore, that a comparison of our paradigm with (frequency-coded) SSVEP BCIs and in particular BCIs with a dynamic trial duration would be more appropriate. We added a respective table to the Discussion.

9. Please provide the details of future direction and possible solutions to continue this topic.

We mention a number of suggestions for future developments and studies at the end of the Discussion.

10. Finally, I suggest authors to sit with English native speaker to improve the writing of proposed work.

We overhauled the manuscript, trying to improve style and fix grammar errors. Should concerns about our writing persist, a few examples for which aspects need improvement would be appreciated.

---

## [Decision Letter · Decision Letter 2]

17 Mar 2022

PONE-D-21-13493R2Instant classification for the spatially-coded BCIPLOS ONE

Dear Dr. Maye,

Thank you for submitting your manuscript to PLOS ONE. After careful consideration, we feel that it has merit but does not fully meet PLOS ONE’s publication criteria as it currently stands. Therefore, we invite you to submit a revised version of the manuscript that addresses the points raised during the review process. Please submit your revised manuscript by May 01 2022 11:59PM. If you will need more time than this to complete your revisions, please reply to this message or contact the journal office at plosone@plos.org. Please include the following items when submitting your revised manuscript:A rebuttal letter that responds to each point raised by the academic editor and reviewer(s). You should upload this letter as a separate file labeled 'Response to Reviewers'.A marked-up copy of your manuscript that highlights changes made to the original version. You should upload this as a separate file labeled 'Revised Manuscript with Track Changes'.An unmarked version of your revised paper without tracked changes. You should upload this as a separate file labeled 'Manuscript'.If applicable, we recommend that you deposit your laboratory protocols in protocols.io to enhance the reproducibility of your results. Protocols.io assigns your protocol its own identifier (DOI) so that it can be cited independently in the future. For instructions see: https://journals.plos.org/plosone/s/submission-guidelines#loc-laboratory-protocols. Additionally, PLOS ONE offers an option for publishing peer-reviewed Lab Protocol articles, which describe protocols hosted on protocols.io. Read more information on sharing protocols at https://plos.org/protocols?utm_medium=editorial-email&utm_source=authorletters&utm_campaign=protocols.

We look forward to receiving your revised manuscript.

Kind regards,

Saeed Mian Qaisar, Ph.D.

Academic Editor

PLOS ONE

Journal Requirements:

Additional Editor Comments:

Dear Author,

Reviewers have now commented on your paper. They are advising that you revise your manuscript. If you are prepared to undertake the work required, I would be please to reconsider my decision.

The reviewer comments can be found at the end of this email or can be accessed online.

Reviewers' comments:

Reviewer's Responses to Questions

**Comments to the Author**

1. If the authors have adequately addressed your comments raised in a previous round of review and you feel that this manuscript is now acceptable for publication, you may indicate that here to bypass the “Comments to the Author” section, enter your conflict of interest statement in the “Confidential to Editor” section, and submit your "Accept" recommendation.

Reviewer #3: (No Response)

2. Is the manuscript technically sound, and do the data support the conclusions?

Reviewer #3: Yes

3. Has the statistical analysis been performed appropriately and rigorously? 

Reviewer #3: Yes

4. Have the authors made all data underlying the findings in their manuscript fully available?

Reviewer #3: No

5. Is the manuscript presented in an intelligible fashion and written in standard English?

Reviewer #3: Yes

6. Review Comments to the Author

Reviewer #3: This paper proposes a new SSVEP-BCI online system based on spatially encoded SSVEP-BCI and dynamic time windows. The system utilizes CCA- and LDA-based SSVEP EEG signals to acquire subjects' intentions. And this paper proposes a dynamic time window based on Bayes' theorem and a special stopping strategy to realize the dynamic change of EEG data length. The research is of great significance to the practical application of BCI technology. However, this article leaves some gaps in the details. The specific questions are as follows. I am personally optimistic about the results of these experiments and look forward to the author's team's follow-up supplements to the paper.

1. Can you describe the workflow of the stimulus interface in detail?

2. We noticed that the experimental results between the best subjects and the worst subjects are very different. What is the reason for this?

3. Please list the promotion ratio in Table 2 to improve the readability of the paper.

4. The serial number is not indicated in Figure 5 to Figure 7 in the picture area.

5. Is there a significant difference between the test results of the offline test and the online test after using the dynamic time window?

6. A study needs to be discussed in this study, such as “A Dynamically Optimized SSVEP Brain–Computer Interface (BCI) Speller[J]. IEEE Transactions on Biomedical Engineering, 2015, 62(6): 1447-1456.”.

7. PLOS authors have the option to publish the peer review history of their article (what does this mean?). If published, this will include your full peer review and any attached files.

Reviewer #3: **Yes: **Erwei Yin

---

## [Author Response · Author response to Decision Letter 2]

8 Apr 2022

1. Can you describe the workflow of the stimulus interface in detail?

The workflow in the training session is described on lines 100–108 and in the online session on lines 114--129 in the section „Stimulation and experimental procedure“. The interface only generated the visual flicker (line 91) and a cue (line 101, Figure 1) which directed the participant’s gaze to the respective target location (numbered 1-5, see Fig. 1) relative to the flicker. We cannot think of any other details about the stimulation or the workflow that might be missing, but we will readily add them if the reviewer makes a more specific request.

2. We noticed that the experimental results between the best subjects and the worst subjects are very different. What is the reason for this?

There are several studies which investigated possible causes for the strong variability in BCI performance across people (e.g., Allison et al., BCI Demographics: How Many (and What Kinds of People) Can Use an SSVEP BCI?, IEEE TNSRE 18(2), 2010), but as far as we know, no factors have been identified. We made an attempt to contribute to this line of research and show in Fig. 7 that the ITR is related to the individual SSVEP response strength of the participants. Of course the next question then is what the reason for this difference in response strength might be. We regret to not have an answer. We nevertheless tried to better understand the observed performance difference and rearranged the bar plot in Fig. 5 by sorting the participants according to their ITR in the online session. The new arrangement suggests that the two subjects with the lowest performance are part of a continuum rather than outliers.

3. Please list the promotion ratio in Table 2 to improve the readability of the paper.

We added the corresponding values to Table 2.

4. The serial number is not indicated in Figure 5 to Figure 7 in the picture area.

This was a problem during the submission process which we solved now, so no changes were necessary in the manuscript.

5. Is there a significant difference between the test results of the offline test and the online test after using the dynamic time window?

We added a comparison of classification accuracies and ITRs to the Results section.

6. A study needs to be discussed in this study, such as “A Dynamically Optimized SSVEP Brain–Computer Interface (BCI) Speller[J]. IEEE Transactions on Biomedical Engineering, 2015, 62(6): 1447-1456.”.

The manuscript discusses this article on line 27–29, lists the results in Table 2 and cites it in the references [3]. Nevertheless, we added a few more details on lines 30--32.

---

## [Editor Report · Decision Letter 3]

12 Apr 2022

Instant classification for the spatially-coded BCI

PONE-D-21-13493R3

Dear Dr. Maye,

We’re pleased to inform you that your manuscript has been judged scientifically suitable for publication and will be formally accepted for publication once it meets all outstanding technical requirements.

Kind regards,

Saeed Mian Qaisar, Ph.D.

Academic Editor

PLOS ONE

Additional Editor Comments (optional):

Dear Authors,

I am pleased to tell you that your work has now been accepted for publication in the PLOS ONE Journal.

Thank you for submitting your work to PLOS ONE Journal.

---

## [Editor Report · Acceptance letter]

20 Apr 2022

PONE-D-21-13493R3 

Instant classification for the spatially-coded BCI 

Dear Dr. Maÿe:

I'm pleased to inform you that your manuscript has been deemed suitable for publication in PLOS ONE. Congratulations! Your manuscript is now with our production department. 

Kind regards, 

on behalf of

Dr. Saeed Mian Qaisar 

Academic Editor

PLOS ONE